# Sensor-Based Hand Gesture Detection and Recognition by Key Intervals

**Yin-Lin Chen [1], Wen-Jyi Hwang [1,\*] 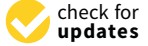, Tsung-Ming Tai [2] and Po-Sheng Cheng [1]**

[1] Department of Computer Science and Information Engineering, National Taiwan Normal University, Taipei 116, Taiwan; 60847080s@ntnu.edu.tw (Y.-L.C.); 60947038s@ntnu.edu.tw (P.-S.C.)
[2] NVIDIA AI Technology Center, Taipei 114, Taiwan; ntai@nvidia.com
\* Correspondence: whwang@ntnu.edu.tw

**Abstract:** This study aims to present a novel neural network architecture for sensor-based gesture detection and recognition. The algorithm is able to detect and classify accurately a sequence of hand gestures from the sensory data produced by accelerometers and gyroscopes. Each hand gesture in the sequence is regarded as an object with a pair of key intervals. The detection and classification of each gesture are equivalent to the identification and matching of the corresponding key intervals. A simple automatic labelling is proposed for the identification of key intervals without manual inspection of sensory data. This could facilitate the collection and annotation of training data. To attain superior generalization and regularization, a multitask learning algorithm for the simultaneous training for gesture detection and classification is proposed. A prototype system based on smart phones for remote control of home appliances was implemented for the performance evaluation. Experimental results reveal that the proposed algorithm provides an effective alternative for applications where accurate detection and classification of hand gestures by simple networks are desired.

**Keywords:** hand gesture detection; hand gesture recognition; neural networks; human–machine interface

## 1. Introduction

The Internet of Things (IoT) is a growing trend with large varieties of consumer and industrial products being connected via the Internet. To operate with these devices, the implementation of a smart human–machine interface (HMI) becomes an important issue. Some HMI systems are based on hand gesture recognition. Although vision-based recognition (VBR) algorithms [1] have been found to be effective, the information of hand gestures is extracted from video sequences captured by cameras. Because VBR systems may record users' life continuously, there are risks of personal information disclosure, which lead to privacy issues [2,3]. In addition, high computational complexities may be required [4] for carrying out the gesture information extraction from video sequences for real-time applications.

Alternatives to VBR algorithms are the sensor-based recognition (SBR) algorithms, which are based on the sensory data produced by devices different from cameras. Privacy-preserved activity recognition can then be achieved. Commonly used sensors include accelerometers [5,6], gyroscopes [4,7], photoplethysmography (PPG) [8], flex sensors [9], electromyography (EMG) [10], and the fusion of these sensors. Although an SBR algorithm may have lower computational complexities, it is usually difficult to extract gestures from sensory data for the subsequent classification in the algorithm. In fact, for some sensory data, a precise gesture extraction can be challenging even by direct visual inspection of the samples because gesture movements may not be easily inferred from their corresponding sensor readings.

The extraction of gestures from sensory data is equivalent to the identification of the start and end points of the gestures. Dedicated sensors, explicit user actions, or special gesture markers are required for the studies in [10–13] for determining the start

and end locations. These methods introduce extra overhead for the gesture detection. The SBR approaches in [14,15] carry out the gesture extraction automatically based on the variances of sensory data. However, because hand movements generally produce large variances, false alarms may be triggered as the unintended gestures are performed. In addition, accurate detection of a sequence of gestures could also be difficult. Deep learning techniques [16] such as long short-term memory (LSTM) and/or convolution neural network (CNN) have been found to be effective for the detection and classification of a sequence of gestures [7,17–19]. However, the techniques operate under the assumptions that the start and end locations of the gesture sequences are available before each individual gesture can be identified.

The goal of this study is to present a novel SBR detection and classification technique for a sequence of hand gestures. The sensors considered in this study are accelerometer and gyroscope. The proposed detection algorithm is automatic so that no dedicated sensors, explicit user actions, or special gesture markers are required. Furthermore, no prior knowledge on the start and end locations of the whole sequence of hand gestures is needed. In the proposed algorithm, each hand gesture in the sequence is regarded as an object. The detection of the object is based on a pair of key intervals. One interval, termed primary key interval (PKI), is located in the first half of the gesture. The other interval, termed secondary key interval (SKI), is in the second half. A gesture is detected when the paired key intervals are identified. The requirement for the detection of the paired key intervals is beneficial for lowering the false alarm rates triggered by unintended gestures.

Furthermore, a simple automatic labelling scheme for the identification of the key intervals is proposed in this study. No manual visual inspection is required. After locating the PKI and SKI, a Gaussian-like function is then adopted for spreading the label values outside the intervals. The label values associated with each sample are the scores indicating the likelihood that the corresponding sample belongs to the key intervals. During the inference process, the scores associated with each sample are then computed for gesture detection. A multitask learning technique [20] is employed for the gesture detection and classification. Because the detection and classification are related tasks, the simultaneous learning of the tasks provides the advantages of a superior generalization and regularization through shared representation, and improved data efficiency as well as fast learning by leveraging auxiliary information offered by the other tasks.

The major contributions of this work are threefold:

1. We present a novel gesture detection and classification algorithm for sensory data based on objects as paired intervals. The algorithm is able to carry out semantic detection with a high detection accuracy and low false positive rate even in the presence of unintended gestures.
2. We propose a simple automatic soft-labelling scheme for the identification of key intervals. The simple labelling scheme is able to facilitate the collection and annotation of training data.
3. We propose a multitask learning algorithm for the simultaneous training for gesture detection and classification. The multitask learning is beneficial for providing superior generalization and regularization for SBR-based training.

The remaining parts of this paper are organized as follows. Section 2 presents the related work to this study. The proposed SBR algorithm is discussed in detail in Section 3. Experimental results of the proposed SBR algorithm are then presented in Section 4. Finally, Section 5 includes some concluding remarks of this work.

## 2. Related Works

The detection of gestures from sensory data can be viewed as an object detection problem. For the computer vision applications, the detection of objects from images is a challenging and fundamental problem. State-of-the-art detection results have been achieved by various deep learning techniques [21]. A common component for some of these approaches is the employment of anchor boxes as detection candidates [22,23]. Anchor

boxes are the boxes with various sizes and aspect ratios. A large set of anchor boxes [24] may be required for accurate detection. Subsequently, high computation overhead is usually introduced for both training and inference.

Alternatives to the anchor-based approaches are to represent each object as a single [25] or multiple keypoints [26,27]. For the technique with a single keypoint, the keypoint of an object is the centre of the bounding box of the object. When an object is represented by a pair or a triplet of keypoints, each keypoint represents the centre or corners of the bounding box. The corresponding object detection operations are equivalent to finding the keypoints of the objects. The need for anchor boxes is then bypassed.

Although the keypoint-based approaches have low computation complexities, they may focus only on the local features of objects for identifying keypoints. By contrast, the proposed work is based on the key intervals for the gesture detection. Global features characterizing the key intervals would then be taken into consideration by the proposed algorithm. A superior detection accuracy can be achieved with a low computation overhead. The joint training for both detection and classification as multitask operations in the proposed algorithm is also beneficial for the effective classification of gestures after detection operations. Similar schemes can also be observed in the study [28] for instance-aware human semantic parsing, where a joint training for different tasks such as keypoint detection, human-part-parsing and body-to-point project were carried out. The corresponding backbone-sharing scheme was found to be effective over its counterparts [29,30] without multitask operations.

### 3. Proposed Algorithm

*3.1. Overview*

Figure 1 shows an example of the operations of the proposed SBR algorithm. For the sake of simplicity, only samples produced from a 3-axis accelerometer were considered for gesture recognition, as shown in the top graph of Figure 1. Furthermore, the system is capable of the detection/classification of two gesture classes. The detection of a single gesture involves the detection of PKI and SKI, which are based on the scores produced by the proposed neural network model. We can observe from the top graph of Figure 1 that the samples in the detection sliding window are served as the inputs to the neural network. For each class, there are separate scores for the detection of PKI and SKI at the output of the neural network.

The second and third graphs of Figure 1 reveal the resulting PKI and SKI scores for the sensory data. The scores are subsequently compared with prespecified thresholds for the detection of PKI and SKI. When a detection of a PKI is followed by a detection of an SKI, and both the PKI and SKI belong to the same gesture class, the detection is matched. Otherwise, the detection is unmatched, and is ignored by the subsequent classification. After the occurrence of a matched detection, the subsequent gesture classification is straightforward. As shown in the second and third graphs of Figure 1, because the key intervals and their associated gesture should belong to the same class, we select the class that both PKI and SKI belong to as the result of the gesture classification.

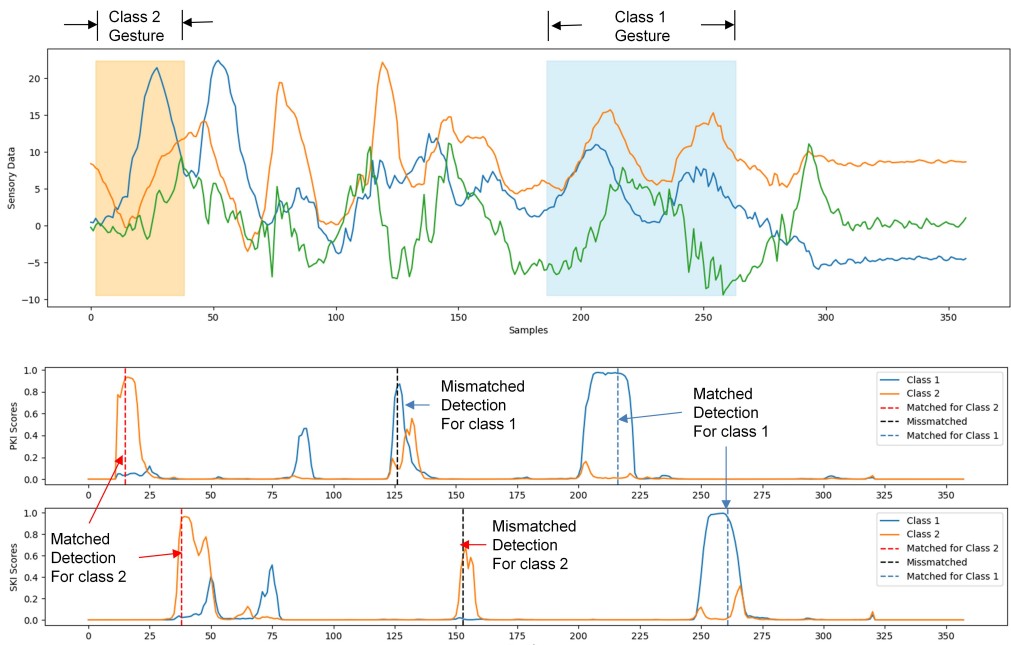

**Figure 1.** An example of the operations of the proposed SBR algorithm. The top graph of this example shows the sensory data from a 3-axis accelerometer. The second and third graphs reveal the PKI and SKI scores produced by the proposed network model, respectively. In the example, a matched detection for class 2 gesture is first identified. A number of mismatched detections are then followed. Finally, a matched detection for class 1 gesture is found.

### 3.2. Proposed Neural Network Model

The architecture of the proposed neural network model for gesture detection and classification is shown in Figure 2. Similar to the study in [28], the proposed architecture contains a backbone, which is shared by three branches. As shown in Figure 2, the lower two branches are used for producing PKI and SKI scores for gesture detection, respectively. The topmost branch is adopted for delivering the classification results. Therefore, the proposed model is a multitask network offering the simultaneous learning of gesture detection and classification.

It can be observed from Figure 2 that a convolution layer containing a convolution (CONV) operation, a Relu-based activation, and a group normalization (GN) [31] form the basic building block for the backbone and the branches. The kernel size, stride size, and number of output channels associated with the CONV operation of each basic building block are also shown in the figure. There are 4 convolution layers (denoted by C1, C2, C3, and C4) in the backbone. In addition to the basic building blocks, the backbone consists of two shortcuts [32] for enhancing the effectiveness of the feature extraction. The convolution layers (denoted by C5, C6, C7, and C8), dense layers (denoted by F1, F2, and F3), average pooling (AP), and softmax are employed in the branches for summarizing the features produced by the backbone for the PKI and SKI detection as well as the gesture classification.

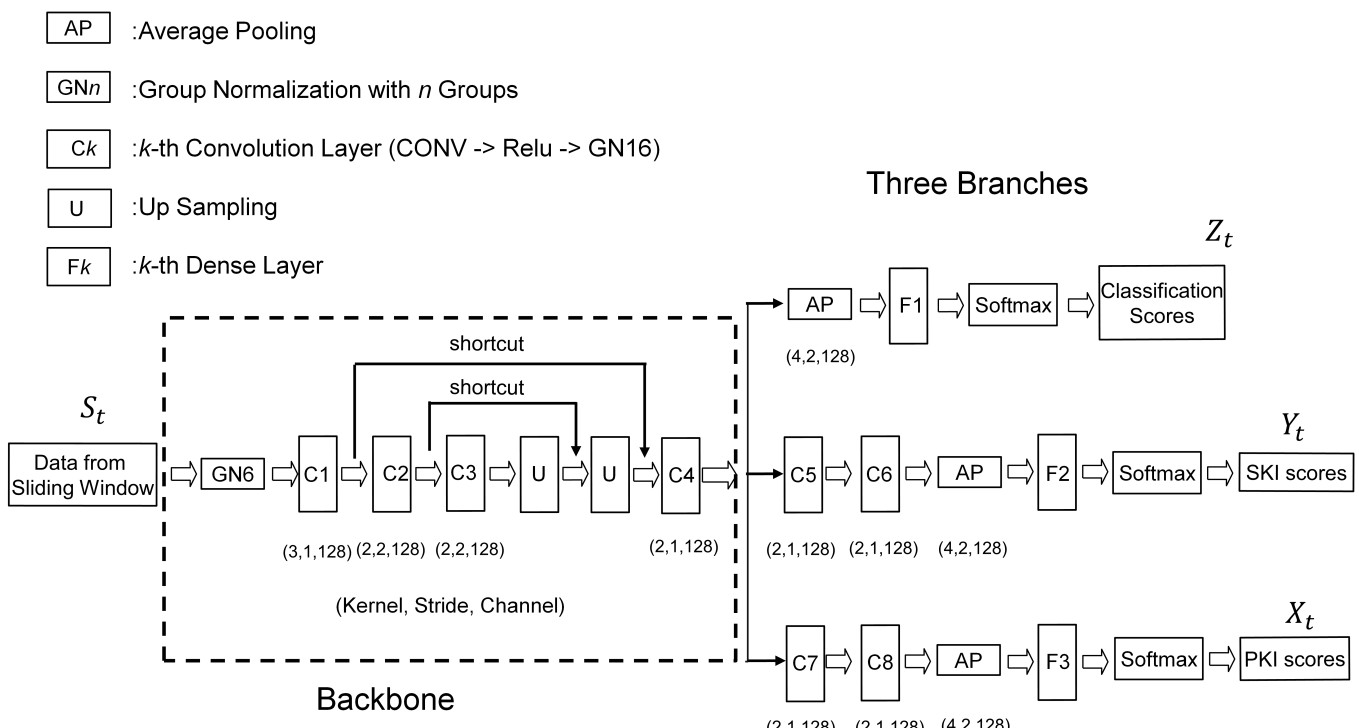

**Figure 2.** The proposed neural network model for gesture detection and classification. The model contains a backbone and three branches. The samples from a sliding window of a sequence $S$ serve as input data. The model then produces PKI scores, SKI scores, and classification scores corresponding to the sliding window.

### 3.3. Training Operations

To facilitate the training and inference operations of the proposed neural network model, a list of commonly used symbols is revealed in Table A1 in Appendix A. Let $S = \{s_1, \ldots, s_L\}$ be an input sensory data sequence for training. Each $s_t \in S$ is the $t$th sample of the input sequence, $t = 1, \ldots, L$, where $L$ is the length of the sequence. Each sample $s_t$ is an $N$-tuple vector, where $N$ is dependent on the sensors adopted for the gesture recognition. For example, $N = 6$ when both 3-axis accelerometer and 3-axis gyroscope are used.

A sliding window operation on the input sequence $S$ is adopted for the training of the neural network. Let $S_t$ be a window of $W$ successive samples from $S$, where the central sample of $S_t$ is $s_t$. In the proposed algorithm, we slide the window $S_t$ with stride size 1. For each $S_t$, three outputs $X_t$, $Y_t$, and $Z_t$ can be obtained from the three branches of the proposed neural network model.

Let $K$ be the number of gesture classes for classification. The $X_t$, $Y_t$, and $Z_t$ are then $(K + 1)$-tuple vectors. Let $X_{t,j}$, $Y_{t,j}$, and $Z_{t,j}$ be the $j$th element of $X_t$, $Y_t$, and $Z_t$, respectively. The $X_{t,j}$, $Y_{t,j}$, and $Z_{t,j}$, $j = 1, \ldots, K$, are the PKI score, SKI score, and classification score associated with class $j$ for the window $S_t$. When $j = K + 1$, the $X_{t,K+1}$, $Y_{t,K+1}$, and $Z_{t,K+1}$ are the scores associated with the background class. Furthermore, because softmax is employed in the proposed network model, we have $X_{t,j} \geq 0$, $Y_{t,j} \geq 0$, $Z_{t,j} \geq 0$, $j = 1, \ldots, K + 1$, and

$$\sum_{j=1}^{K+1} X_{t,j} = 1, \quad \sum_{j=1}^{K+1} Y_{t,j} = 1, \quad \sum_{j=1}^{K+1} Z_{t,j} = 1. \tag{1}$$

For each window $S_t$, let $F_t$, $G_t$ and $H_t$ be the ground truth of $X_t$, $Y_t$, and $Z_t$, respectively. Therefore, they are also $(K + 1)$-tuple vectors. Based on the training sequence $S$, the assignments of ground truth values to $F_t$, $G_t$, and $H_t$ for each window $S_t$ are regarded as the labelling process for the training operation.

The proposed labelling process operates under the assumption that the start location and end location for each gesture in the training sequence $S$ are known. A simple algorithm is then proposed to find the PKI and SKI associated with the gesture numerically. Based on the PKI and SKI associated with each gesture, the $F_t$, $G_t$, and $H_t$ for each window $S_t$ are then computed.

Figure 3 shows an example of the labelling process for a training sequence $S$ containing only a single gesture. Let $P_s$ and $P_f$ be the locations of the start and final samples of the gesture, respectively. Let $u_c$ and $R$ be the centroid and radius of the gesture, respectively. That is,

$$u_c = \frac{P_f + P_s}{2}, \ R = \frac{P_f - P_s}{2}. \tag{2}$$

Therefore, the set of indices of the gesture, denoted by $\mathcal{I}_R$, is given by

$$\mathcal{I}_R = \{t : u_c - R \leq t \leq u_c + R\}. \tag{3}$$

Let $u_{\mathrm{PKI}}$ and $u_{\mathrm{SKI}}$ be the centroid of the PKI and SKI, respectively. In this study, $u_{\mathrm{PKI}}$ and $u_{\mathrm{SKI}}$ were given by

$$u_{\mathrm{PKI}} = u_c - 0.4R, \ u_{\mathrm{PKI}} = u_c + 0.4R. \tag{4}$$

Both PKI and SKI have the same length, denoted by $I$, where

$$I = rR, \tag{5}$$

where $0 < r < 1$ is a positive constant. In this study, we termed $r$ the key interval-length-to-gesture-radius (ITR) ratio. For the example shown in Figure 3, the ITR ratio $r = 0.3$. Let $\mathcal{I}_{\mathrm{PKI}}$ and $\mathcal{I}_{\mathrm{PKI}}$ be the set of indices belong to PKI and SKI of the gesture, respectively. They are then given by

$$\mathcal{I}_{\mathrm{PKI}} = \{t : u_{\mathrm{PKI}} - \frac{I}{2} \leq t \leq u_{\mathrm{PKI}} + \frac{I}{2}\}, \tag{6}$$

$$\mathcal{I}_{\mathrm{SKI}} = \{t : u_{\mathrm{SKI}} - \frac{I}{2} \leq t \leq u_{\mathrm{SKI}} + \frac{I}{2}\}. \tag{7}$$

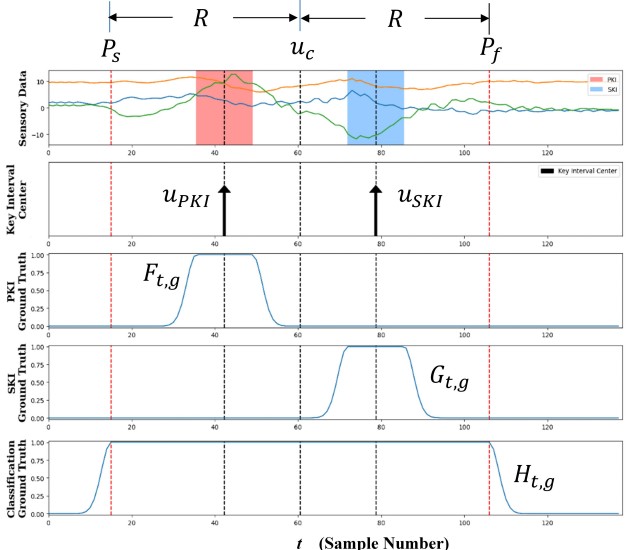

**Figure 3.** An example of labelling process for a training sequence $S$ containing only a single gesture. The $P_f$ and $P_s$ of the top graph are the locations of the start and final samples of the gesture, respectively. The $u_c$ of the top graph is the centroid of the gesture. The second graph marks the locations of $u_{\mathrm{PKI}}$ and $u_{\mathrm{SKI}}$ as black arrows. The third, fourth, and the bottom graphs reveal the Gaussian-like ground truth of the gesture for PKI score, SKI score, and classification score, respectively.

Assume class $g$ is the ground truth of the gesture. Furthermore, let $F_{t,j}$, $G_{t,j}$ and $H_{t,j}$ be the ground truth of the PKI score, SKI score, and classification score associated with class $j$ for the window $S_t$. In this study, a Gaussian-like function was adopted for the assignments of $F_{t,g}$, $G_{t,g}$, and $H_{t,g}$ as follows.

$$F_{t,g} = \begin{cases} 1 & \text{, if } t \in \mathcal{I}_{\text{PKI}}, \\ e^{-(|t-u_{\text{PKI}}|-\frac{I}{2})^2/2\sigma^2} & \text{, otherwise.} \end{cases} \tag{8}$$

$$G_{t,g} = \begin{cases} 1 & \text{, if } t \in \mathcal{I}_{\text{SKI}}, \\ e^{-(|t-u_{\text{SKI}}|-\frac{I}{2})^2/2\sigma^2} & \text{, otherwise.} \end{cases} \tag{9}$$

$$H_{t,g} = \begin{cases} 1 & \text{, if } t \in \mathcal{I}_{R}, \\ e^{-(|t-u_c|-\frac{R}{2})^2/2\sigma^2} & \text{, otherwise.} \end{cases} \tag{10}$$

Examples of $F_{t,g}$, $G_{t,g}$, and $H_{t,g}$ are shown in the third, fourth, and bottom graphs of Figure 3, respectively. After the ground truth of the PKI score, SKI score, and classification score associated with class $g$ are determined, we then compute the ground truth of the scores of the other classes $j$, $j \neq g$, as

$$F_{t,j} = \begin{cases} 0 & \text{, } j \neq g, j = 1, ..., K, \\ 1 - F_{t,g} & \text{, } j = K+1. \end{cases} \tag{11}$$

$$G_{t,j} = \begin{cases} 0 & \text{, } j \neq g, j = 1, ..., K, \\ 1 - G_{t,g} & \text{, } j = K+1. \end{cases} \tag{12}$$

$$H_{t,j} = \begin{cases} 0 & \text{, } j \neq g, j = 1, ..., K, \\ 1 - H_{t,g} & \text{, } j = K+1. \end{cases} \tag{13}$$

Based on the assignment, it can be easily shown that

$$\sum_{j=1}^{K+1} F_{t,j} = 1, \quad \sum_{j=1}^{K+1} G_{t,j} = 1, \quad \sum_{j=1}^{K+1} H_{t,j} = 1. \tag{14}$$

The constraints in Equation (14) for the ground truth of the scores are consistent with those in Equation (1) for the scores produced by the proposed network model.

Let $J$ be the loss function for the training of the proposed network model for a training sequence $S$. In this study, $J$ was given by

$$J = A + B + C, \tag{15}$$

where $A$, $B$, and $C$ are the loss function for the PKI score, SKI score, and classification scores produced by the proposed neural network model from the training sequence $S$, respectively. A variant of the focal loss function [24] was adopted for the functions $A$, $B$, and $C$. That is,

$$A = \sum_t A_t, \ B = \sum_t B_t, \ C = \sum_t C_t, \tag{16}$$

where $A_t$, $B_t$, and $C_t$ are the losses measured for the window $S_t$. They are given by

$$A_t = \sum_{j=1}^{K+1} A_{t,j}, \ B_t = \sum_{j=1}^{K+1} B_{t,j}, \ C_t = \sum_{j=1}^{K+1} C_{t,j}, \tag{17}$$

where $A_{t,j}$, $B_{t,j}$, and $C_{t,j}$ are the losses due to the PKI score $X_{t,j}$, SKI score $Y_{t,j}$, and classification score $Z_{t,j}$, respectively. They are evaluated by

$$A_{t,j} = \begin{cases} -(1 - X_{t,j})^\alpha \log(X_{t,j}) & \text{, if } t \in \mathcal{I}_{\text{PKI}}, \\ -(1 - F_{t,j})^\beta (X_{t,j})^\alpha \log(1 - X_{t,j}) & \text{, otherwise.} \end{cases} \tag{18}$$

$$B_{t,j} = \begin{cases} -(1 - Y_{t,j})^{\alpha} \log(Y_{t,j}) & \text{, if } t \in \mathcal{I}_{\text{SKI}}, \\ -(1 - G_{t,j})^{\beta}(Y_{t,j})^{\alpha} \log(1 - Y_{t,j}) & \text{, otherwise.} \end{cases} \tag{19}$$

$$C_{t,j} = \begin{cases} -(1 - Z_{t,j})^{\alpha} \log(Z_{t,j}) & \text{, if } t \in \mathcal{I}_{R}, \\ -(1 - H_{t,j})^{\beta}(Z_{t,j})^{\alpha} \log(1 - Z_{t,j}) & \text{, otherwise.} \end{cases} \tag{20}$$

The loss functions are variants of the focal loss functions proposed in [24], where parameters $\alpha > 0$ and $\beta > 0$ should be prespecified before the training process.

Although the loss function $J$ in Equation (15) is computed only for a single input training sequence $S$, it can be easily extended for a training set containing multiple training sequences. This is accomplished by simply evaluating $J$ for each individual training sequence in the set. The overall loss for the training set is then the sum of the $J$ for each training sequence.

We can also observe from Equation (15) that the losses $A$ and $B$ are the losses for detection, and $C$ is for classification. Therefore, the proposed training algorithm can be viewed as a multitask learning technique for the simultaneous learning of detection and classification, which are the related tasks sharing the same backbone network in the proposed neural network shown in Figure 2. The proposed technique therefore provides the advantages of a superior generalization and regularization through shared representation from the backbone network.

*3.4. Inference Operations*

For the inference operations, the input sequence $S$ could contain more than one hand gesture. Furthermore, the start and end locations of the samples in each gesture are not known for the inference operations. As shown in Figure 4, the inference process first activates the detector of key intervals, which evaluates the PKI and SKI scores for the current sliding window $S_t$. The scores are then compared with a threshold, denoted by $T$, for the detection of key intervals. If a key interval is detected, a matching process for the key intervals is then initiated to determine whether an occurrence of a gesture is found or not. When a detected PKI is followed by a detected SKI, and both PKI and SKI belong to the same gesture class, we then say that a gesture is detected and classified.

For example, suppose $t_1$ and $t_2$ are the indices where PKI and SKI are detected. Therefore, the highest PKI score in $X_{t_1}$ and the highest SKI score in $Y_{t_2}$ for nonbackground classes should be higher than the threshold $T$. That is, let

$$j^* = \operatorname*{argmax}_{1 \le j \le K+1} X_{t_1,j}, \ \ j^{**} = \operatorname*{argmax}_{1 \le j \le K+1} Y_{t_2,j}. \tag{21}$$

It then follows that

$$X_{t_1,j^*} > T, \ \ Y_{t_2,j^{**}} > T, \tag{22}$$

where $j^* \ne K + 1$, $j^{**} \ne K + 1$, and class $K + 1$ is the background class. Suppose no PKI or SKI detections occur between $t_1$ and $t_2$. A matched detection then occurs when $j^* = j^{**}$. In this case, a gesture is detected. In addition, the gesture is classified as class $j^{**}$.

To further enhance the robustness of the proposed algorithm, additional constraints may be imposed. In this study, we required that $t_{\min} < t_1 - t_2 < t_{\max}$, where $t_{\min}$ and $t_{\max}$ are the minimum time and maximum time between a pair of matched PKI and SKI detections, respectively. This would reduce the possibilities of false matches due to inaccurate PKI or SKI detections. As revealed in Figure 4, a postprocessing operation is employed for enforcing the constraints for the gesture detection.

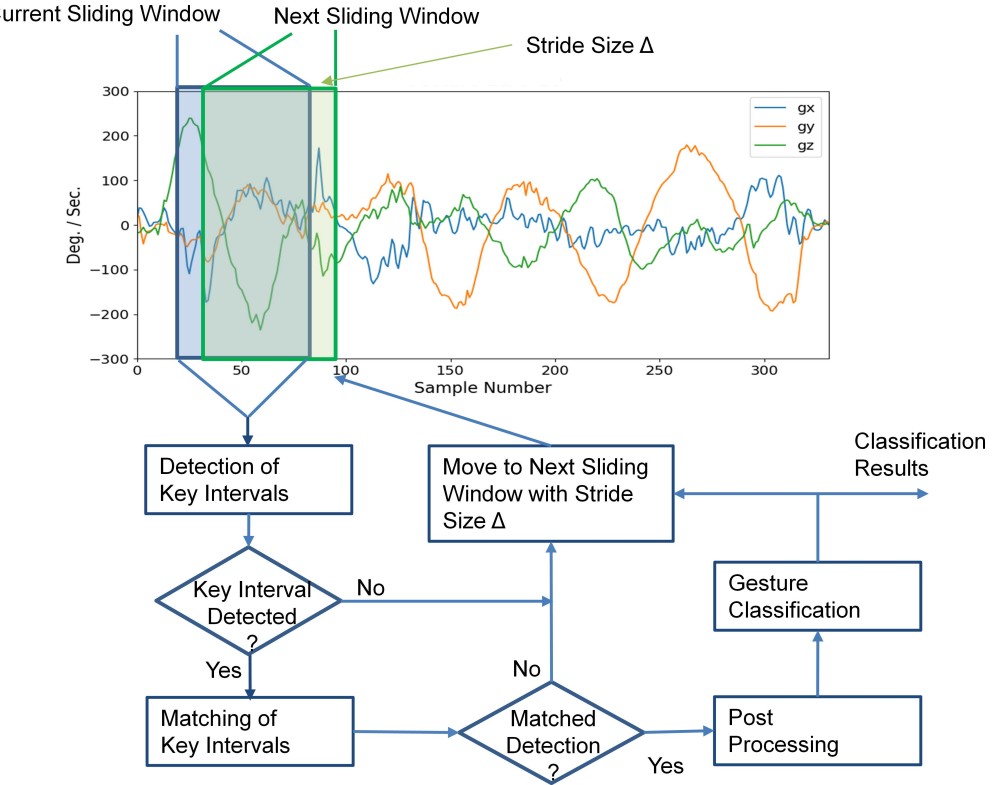

**Figure 4.** The inference operations of the proposed neural network for an input sequence. There are three major operations for each sliding window of the input sequence: detection of key intervals, key interval matching, and postprocessing. The same operations are repeated for each sliding window with stride size Δ until the final window of the input sequence is reached.

For each sliding window $S_t$ from input sequence $S$, the operations of key interval detection, key interval matching, and postprocessing are carried out sequentially, where the next sliding window is obtained from the current one with stride size Δ. The same operations are repeated for each sliding window until the final window of $S$ is processed. Algorithm 1 summarizes the detailed operations for the inference process. The sets $\mathcal{C}$ and $\mathcal{T}$ returned by Algorithm 1 consist of the gesture class and the occurrence time of the detected gestures, respectively.

An important fact in the proposed inference algorithm is that the classification scores in $Z_t$ are not required for the gesture classification. We include $Z_t$ only in training process, where it is adopted for the computation of $C$ in Equation (16) for loss function $J$ in Equation (15). We can view the incorporation of $C$ for the loss evaluation as a regularization scheme for the detection. The backbone of the proposed neural network is then able to provide features best for both the detection and classification. Therefore, $Z_t$ may not be needed. Only the KPI scores $X_t$ and $Y_t$ are involved in the gesture classification.

---

**Algorithm 1** Proposed Gesture Inference Procedure

---

**Require:** Input sequence $S$.
**Require:** A trained neural network model.
**Require:** Threshold $T$ for detection of PKI and SKI.
**Require:** Stride size $\Delta > 0$.
**Require:** $\mathcal{T} = \varnothing, \mathcal{C} = \varnothing$.
**Require:** PKI_Detected $\leftarrow$ false.
**Require:** SKI_Detected $\leftarrow$ false.

 1:  $S_t$ is the first window of $S$.
 2: **repeat**
 3:     Find $X_t$ and $Y_t$ from $S_t$ by the trained neural network.
 4:     Compute $j^* \leftarrow \mathrm{argmax}_{1 \le j \le K+1} X_{t,j}$.
 5:     **if** $(j^* \ne K+1)$ **and** $(X_{t,j^*} > T)$ **then**                                      ▷ PKI detection
 6:         PKI_Detected $\leftarrow$ true.
 7:         PKI_Detected_Time $\leftarrow t$.
 8:         PKI_Class $\leftarrow j^*$.
 9:     **end if**
10:     Compute $j^{**} \leftarrow \mathrm{argmax}_{1 \le j \le K+1} Y_{t,j}$.
11:     **if** $(j^{**} \ne K+1)$ **and** $(Y_{t,j^{**}} > T)$ **then**                                 ▷ SKI detection
12:         SKI_Detected $\leftarrow$ true.
13:         SKI_Detected_Time $\leftarrow t$.
14:         SKI_Class $\leftarrow j^{**}$.
15:     **end if**
16:     **if** (PKI_Detected == true) **and** (SKI_Detected == true) **then**
17:         **if** (PKI_Class == SKI_Class) **then**                          ▷ Matching of key intervals
18:             $\delta \leftarrow$ SKI_Detected_Time $-$ PKI_Detected_Time.
19:             **if** $(t_{\min} < \delta < t_{\max})$ **then**                        ▷ Postprocessing
20:                 Gesture_Class $\leftarrow$ SKI_Class.
21:                 $\mathcal{C} \leftarrow \mathcal{C} \cup$ Gesture_Class.
22:                 $\mathcal{T} \leftarrow \mathcal{T} \cup$ (SKI_Detected_Time, PKI_Detected_Time).
23:                 PKI_Detected $\leftarrow$ false.
24:                 SKI_Detected $\leftarrow$ false.
25:             **end if**
26:         **end if**
27:     **end if**
28:     $t \leftarrow t + \Delta$.
29:     Form new window $S_t$ based on new $t$.
30: **until** Final window of $S$ is processed.
31: **return** $\mathcal{C}, \mathcal{T}$

---

## 4. Experimental Results

This section presents some experimental results of the proposed algorithm. In the experiments, all the gesture sequences for training and testing were acquired by a smartphone equipped with an accelerometer and a gyroscope. The sensors were capable of measuring acceleration and angular velocity in three orthogonal axes, respectively. Therefore, the dimension of each sample $s_t$ was $N = 6$. The size of windows $S_t$ was $W = 50$. The sampling rate was 50 samples/s. For the inference operations, the stride size was $\Delta = 1$. The smart phones adopted for the experiments were a Samsung Galaxy S8 and an HTC ONE M9. A Java-based app was built on the smartphones for gesture capturing and delivery.

As shown in Figure 5, there were five foreground gesture classes (i.e., $K = 5$) for the detection and classification. Samples of foreground gestures were recorded as training data for the training of the proposed network model. Let $M_{\mathrm{TR}}$ be the number of the gestures in the training set. In the experiments, training sets with $M_{\mathrm{TR}} = 180, 320, 450, 590, 750$ gestures were considered. In this way, the impact on the performance of the proposed algorithm for different sizes of training sets could be evaluated. Samples of gestures for each foreground class in the training set are shown in Figure 6. The training operations are implemented by Keras [33].

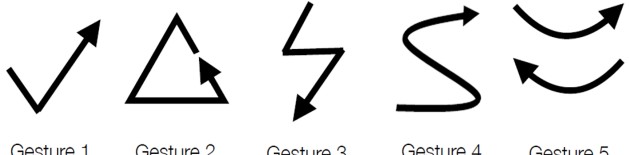

**Figure 5.** The five foreground gesture classes considered in both training and testing sets.

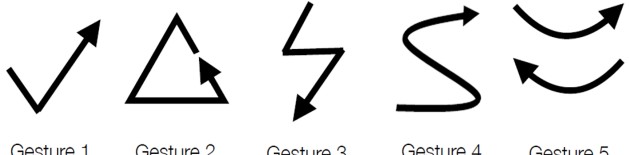

**Figure 6.** Sensory sequences of foreground gestures (i.e., Gesture 1, Gesture 2, Gesture 3, Gesture 4, Gesture 5 defined in Figure 5) in the training set. The boundary of each gesture is marked in the corresponding sequences. Left column: sensory sequence in three orthogonal axes produced by accelerometer; right column: sensory sequence in three orthogonal axes produced by gyroscope, (**a**) Gesture 1, (**b**) Gesture 2, (**c**) Gesture 3, (**d**) Gesture 4, and (**e**) Gesture 5.

The testing set for performance evaluation was different from the training sets. In addition, there were gestures in the test sequences which did not belong to the foreground

gesture classes defined in Figure 5. These gestures were termed background gestures in this study. Figure 7 shows the background gesture classes considered in the experiments. These background classes contained only simple gestures which could be parts of the foreground gesture. In this way, the effectiveness of the proposed algorithm for ignoring background gestures and detecting foreground gestures could be evaluated. Let $M_{TE}$ be the number of gestures in the testing set.

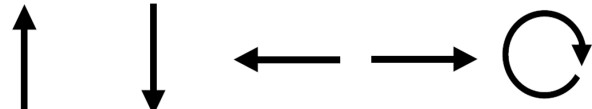

**Figure 7.** The background gesture classes considered in the testing set.

In the experiments, there were 1503 sequences in the testing set. Each sequence may contain two or more hand gestures. The total number of gestures in the testing set was $M_{TE} = 3283$. Among these testing gestures, 1617 gestures were foreground gestures, and 1666 gestures were background gestures. The initial orientation of the smartphone for data acquisition of both training and testing sequences was the portrait orientation. All our experimental results were evaluated on the same testing set.

The network model adopted by the experiments is shown in Figure 2. There were eight convolution layers and three dense layers in the model. Table 1 shows the number of weights in each layer, the number of weights for the backbone, and the branches in the model. We can see from Table 1 that the backbone had the largest number of weights compared with the branches in the model. This was because the backbone contained four convolution layers for effective feature extraction. A fewer number of convolution layers were needed in the branches. The total number of weights in the model was only 282,142, which is the sum of the number of weights for the backbone and branches.

**Table 1.** The size of the proposed neural network model. The total number of weights in the model is 282,142.

| Layer Name | GN6 | C1 | Backbone C2 | C3 | C4 | Branch 1 F1 | C5 | Branch 2 C6 | F2 | C7 | Branch 3 C8 | F3 |
|---|---|---|---|---|---|---|---|---|---|---|---|---|
| Weight size | 12 | 2560 | 33,024 | 33,024 | 33,024 | 16,134 | 33,024 | 33,024 | 16,134 | 33,024 | 33,024 | 16,134 |
| Subtotal | | | 101,644 | | | 16,134 | | 82,182 | | | 82,182 | |

The separate evaluations of gesture detection and classification are first presented. The detection performance was evaluated by the receiver operating characteristic (ROC) curve [34], which is a curve formed by pairs of true positive rate (TPR) and false positive rate (FPR) for various threshold values $T$ for gesture detection. The TPR is defined as the number of correctly detected foreground gestures divided by the total number of foreground gestures in the testing set. The FPR is defined as the number of falsely detected background gestures divided by the total number of background gestures in the testing set. The neural network for the experiment was trained by a set with $M_{TR} = 750$ gestures. The ITR ratio $r$ in Equation (5) was set to 0.3. That is, the key interval length $I$ was only 30% of the gesture radius $R$. Figure 8 shows the corresponding ROC curve, where the corresponding area under the ROC (AUROC) is 0.954. Therefore, with a low FPR, a high TPR can be achieved by the proposed algorithm.

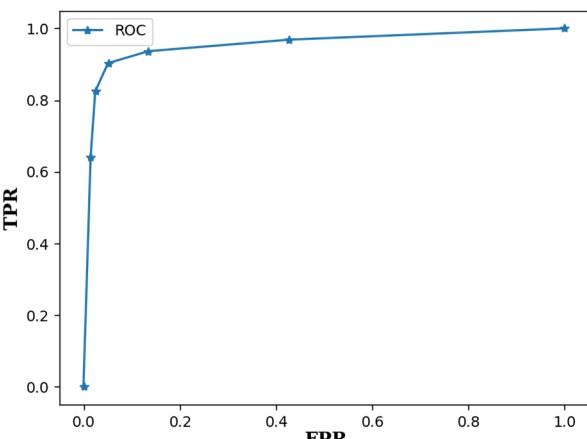

**Figure 8.** The ROC curve of the proposed algorithm for the testing set. The neural network was trained by a set with size $M_{TR} = 750$ and ITR ratio $r = 0.3$. The corresponding AUROC is 0.954.

To further evaluate the detection performance of the proposed algorithm, we compared the AUROC of the proposed algorithm with that of the CornerNet [26] for various sizes $M_{TR}$ of training sets. For the proposed algorithm, we set the ITR ratio $r = 0.3$. The CornerNet is based on keypoints for object detection. The corresponding results are revealed in Table 2. All the training set contained the same number of classes $K = 5$. We can see from Table 2 that the proposed algorithm has a high AUROC even when the size of the training set is small. Furthermore, the AUROC becomes higher as the size of the training set increases. Based on the same training set, the proposed algorithm outperforms the CornerNet in AUROC for detection. These results justify the employment of key intervals for the detection of hand gestures.

**Table 2.** The AUROC performance of the proposed algorithm and CornerNet [26] for gesture detection for various sizes of training sets. The ITR ratio for the proposed algorithm was $r = 0.3$.

| Training Set Size $M_{TR}$ | 180 | 320 | 450 | 590 | 750 |
|---|---|---|---|---|---|
| Proposed | 0.920 | 0.938 | 0.940 | 0.941 | 0.954 |
| CornerNet [26] | 0.845 | 0.862 | 0.875 | 0.878 | 0.882 |

The impact of the ITR ratio $r$ on the performance of the proposed algorithm is revealed in Table 3 for various training set sizes $M_{TR}$. As $r$ increases, it can be observed from Table 3 that the performance of the proposed algorithm can be improved. In fact, when ITR ratio $r$ is above 0.2, the proposed algorithm is able to achieve an AUROC above 0.91 for all the training set sizes considered in the experiments. On the contrary, when $r = 0.1$, the AUROC may be below 0.9 when $M_{TR} = 180$. Therefore, larger values of ITR ratio $r$ are beneficial for improving the accuracy and robustness of the proposed algorithm.

**Table 3.** The AUROC performance of the proposed algorithm with various training set sizes $M_{TR}$ and various values of ITR ratio $r$. The AUROC measurements were based on the same testing set.

| Training Set Size $M_{TR}$ | 180 | 320 | 450 | 590 | 750 |
|---|---|---|---|---|---|
| $r = 0.1$ | 0.857 | 0.892 | 0.910 | 0.921 | 0.926 |
| $r = 0.2$ | 0.915 | 0.917 | 0.923 | 0.935 | 0.938 |
| $r = 0.3$ | 0.920 | 0.938 | 0.940 | 0.941 | 0.954 |
| $r = 0.4$ | 0.939 | 0.940 | 0.943 | 0.944 | 0.953 |
| $r = 0.5$ | 0.941 | 0.952 | 0.955 | 0.955 | 0.957 |

We next considered the classification performance for the correctly detected foreground gestures. Table 4 shows the corresponding confusion matrix of the proposed algorithm with $r = 0.3$. The corresponding neural network was trained by a set with size $M_{TR} = 750$. Each cell in the confusion matrix represents the percentage of the gesture in the corresponding row classified as the gesture in the corresponding column. Let $Q_i$ be the classification accuracy of gesture class $i$, which is defined as the number of gestures in class $i$ that are correctly classified divided by the total number of gestures in class $i$. Therefore, $Q_i$ is the value of the cell in the $i$th column and $i$th row of the confusion matrix. As revealed in Table 4, the proposed algorithm attains a high classification accuracy $Q_i$ for all the gesture classes.

**Table 4.** The confusion matrix on the correctly detected foreground gestures for the proposed algorithm with $M_{TR} = 750$ and $r = 0.3$. The cell located at row $i$ and column $j$ of the matrix represents the percentage in which Gesture $i$ is classified as Gesture $j$.

|         | Gest. 1 | Gest. 2 | Gest. 3 | Gest. 4 | Gest. 5 |
|---------|---------|---------|---------|---------|---------|
| Gest. 1 | 100.00  | 0.00    | 0.00    | 0.00    | 0.00    |
| Gest. 2 | 0.50    | 99.25   | 0.00    | 0.25    | 0.00    |
| Gest. 3 | 0.00    | 0.00    | 100.00  | 0.00    | 0.00    |
| Gest. 4 | 0.00    | 0.00    | 0.45    | 99.55   | 0.00    |
| Gest. 5 | 0.00    | 0.53    | 0.00    | 0.00    | 99.47   |

The proposed algorithm is also able to operate in conjunction with other classification algorithms. In these cases, the proposed algorithm serves only as the gesture detector. Existing gesture classification techniques such as support vector machine (SVM) [6], LSTM [7], bidirectional LSTM (Bi-LSTM) [17], CNN [18], and Residual PairNet [19] can then be adopted to classify the detected gestures. Table 5 shows the classification accuracies of these classification algorithms. For comparison purpose, the proposed algorithm for both detection and classification was also considered in Table 5. We can see from the table that the proposed algorithm outperforms the other algorithms for classification. This is because the joint training of both detection and classification in the proposed algorithm is beneficial for simultaneous detection and classification. That is, when PKI and SKI are matched, the corresponding class is the gesture class of the detected gesture. No other additional efforts are needed for the classification.

**Table 5.** Classification accuracies (in percentage) of various algorithms, where $Q_i$ is the classification accuracy of gesture class $i$. We define $Q_i$ as the number of gestures in class $i$ that are correctly classified divided by the total number of gestures in class $i$.

|                      | $Q_1$  | $Q_2$  | $Q_3$  | $Q_4$  | $Q_5$  |
|----------------------|--------|--------|--------|--------|--------|
| Proposed             | 100.00 | 99.25  | 100.00 | 99.55  | 99.47  |
| SVM [6]              | 90.38  | 87.66  | 86.06  | 73.76  | 89.42  |
| LSTM [7]             | 92.89  | 99.25  | 95.82  | 100.00 | 99.47  |
| Bi-LSTM [17]         | 87.87  | 98.49  | 100.00 | 99.55  | 99.47  |
| CNN [18]             | 94.14  | 98.74  | 98.95  | 99.55  | 99.47  |
| Residual PairNet [19] | 89.12  | 98.99  | 97.56  | 100.00 | 99.47  |

In addition to the separate evaluation of detection and classification, the combined evaluation was also considered in this study. To carry out the evaluation, we considered a sequence of gestures as a string of characters, where each character corresponds to a gesture. The alphabet of the characters was the set of all the foreground gesture classes. The evaluation of the classification results of a gesture sequence was then based on the edit distance [35] between two strings, where one string corresponds to the ground truth of the sequence, and the other is the classification results of the sequence.

In the edit distance between two strings, the displacements, deletions, and insertions of characters from one string to the other are taken into consideration [35]. A displacement

corresponds to the misclassification of one foreground gesture to another foreground gesture. A deletion implies a misdetection of one foreground gesture. An insertion would be the results of the false detection of a background gesture as a foreground one, or the multiple detections of a single foreground gesture. Let $E$ be the edit distance between two gesture sequences: one is the ground truth sequence, and the other is its prediction by the proposed algorithm. Furthermore, let $U$ be the length of the ground truth of the gesture sequence (in number of gestures). We then defined the edit distance accuracy (EDA) as

$$\text{EDA} = 1 - (E/U). \tag{23}$$

Based on the definition, the EDA with highest accuracy is EDA = 1.0. As an example, consider a gesture sequence $S = \{\text{Gesture 2, Gesture 4, Gesture 1}\}$. After the gesture detection and classification, suppose the outcome is $S_1 = \{\text{Gesture 2, Gesture 3, Gesture 5, Gesture 1}\}$. Because $S$ actually contains three gestures, the length of the ground truth is $U = 3$. There exists one displacement and one insertion between $S$ and $S_1$. The edit distance is $E = 2$. From Equation (23), the EDA is $1/3$.

Figure 9 shows the average EDA of the proposed algorithm for various threshold values for detection $T$. The average EDA was measured on the gesture sequences in the testing set. For comparison purpose, the average EDA of the proposed algorithm trained without regularization was also considered in Figure 9. Recall that the regularization was imposed by including the network branch for classification scores in the training process. Nevertheless, the branch and the scores were not used for inference. Therefore, it would be possible to remove from the training the network branch used for producing classification scores. However, it can be observed from Figure 9 that regularization was beneficial for improving the network performance. In fact, it outperformed its counterpart without regularization for all the thresholds considered in this experiment. In particular, when $T = 0.5$, the EDA values of the proposed algorithm with and without regularization were 0.879 and 0.821, respectively. An improvement in EDA by 0.058 was observed. These results justify the employment of regularization for the proposed algorithm.

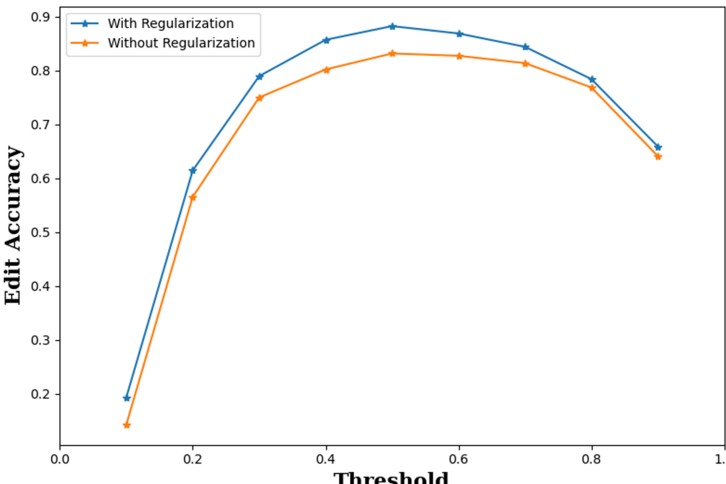

**Figure 9.** The average EDA of the proposed algorithm with and without regularization. We set $M_{\text{TR}} = 750$, $\Delta = 1$ and $r = 0.3$ for the implementation of the proposed algorithm.

Another advantage is that the proposed algorithm may not be sensitive to the selection of thresholds. It can be observed from Figure 9 that the average EDA of the proposed algorithm is higher than 0.8 for $T$ in the range of 0.4 to 0.8. The robustness of the proposed algorithm would be beneficial for providing a reliable performance for gesture detection and classification without the requirement for an elaborate search on the threshold.

Although the experiments considered above were based on an inference procedure with stride size $\Delta = 1$, larger stride sizes can also be considered at the expense of a lower

EDA performance. Table 6 shows the average EDA of the proposed algorithm for various stride sizes for inference. Two thresholds $T = 0.5$ and $T = 0.7$ were considered for the detection of PKI and SKI. It can be observed from Table 6 that it is possible to maintain an average EDA above 0.8 even for stride size $\Delta = 4$. Furthermore, inference operations based on stride sizes $\Delta = 1$ and $\Delta = 2$ attained the same average EDA performance. In particular, when $T = 0.5$, the average EDA was 0.879 for both $\Delta = 1$ and 2. This implies the number of sliding windows computed for the gesture detection and classification can be reduced by half without sacrificing the performance.

**Table 6.** The average EDA performance of the proposed algorithm for various stride sizes for inference. The ITR ratio for the proposed algorithm was $r = 0.3$.

| Stride Size $\Delta$ | 1 | 2 | 3 | 4 | 5 | 6 |
|---|---|---|---|---|---|---|
| $T = 0.5$ | 0.879 | 0.879 | 0.851 | 0.834 | 0.802 | 0.791 |
| $T = 0.7$ | 0.844 | 0.844 | 0.820 | 0.812 | 0.789 | 0.772 |

Finally, some examples for gesture detection and classification are revealed in Figure 10. There were three test sequences considered in the experiments. Each sequence was a mixture of foreground gestures and background samples/gestures. To visualize the effectiveness of the proposed algorithm, the test sequences shown in Figure 10 were randomly selected from the testing set adopted in this study. From Figure 10, we see that the foreground gestures can still be effectively identified even with the presence of background gestures. Please note that the background gestures defined in Figure 7 were the simple gestures constituting other unintended gestures in real-life applications. Therefore, the avoidance of unexpected triggering of these background gestures is beneficial for an accurate gesture detection and classification. All these results show the effectiveness of the algorithm.

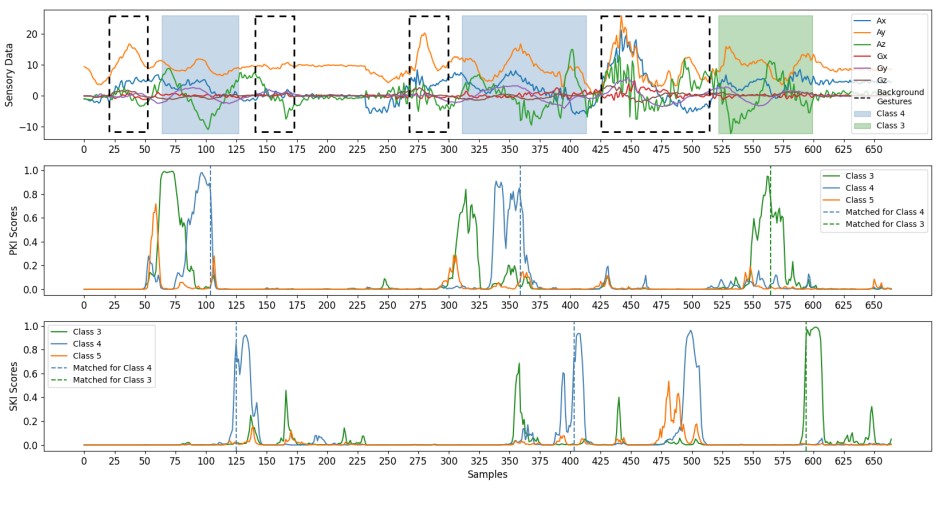

(a)

**Figure 10.** *Cont.*

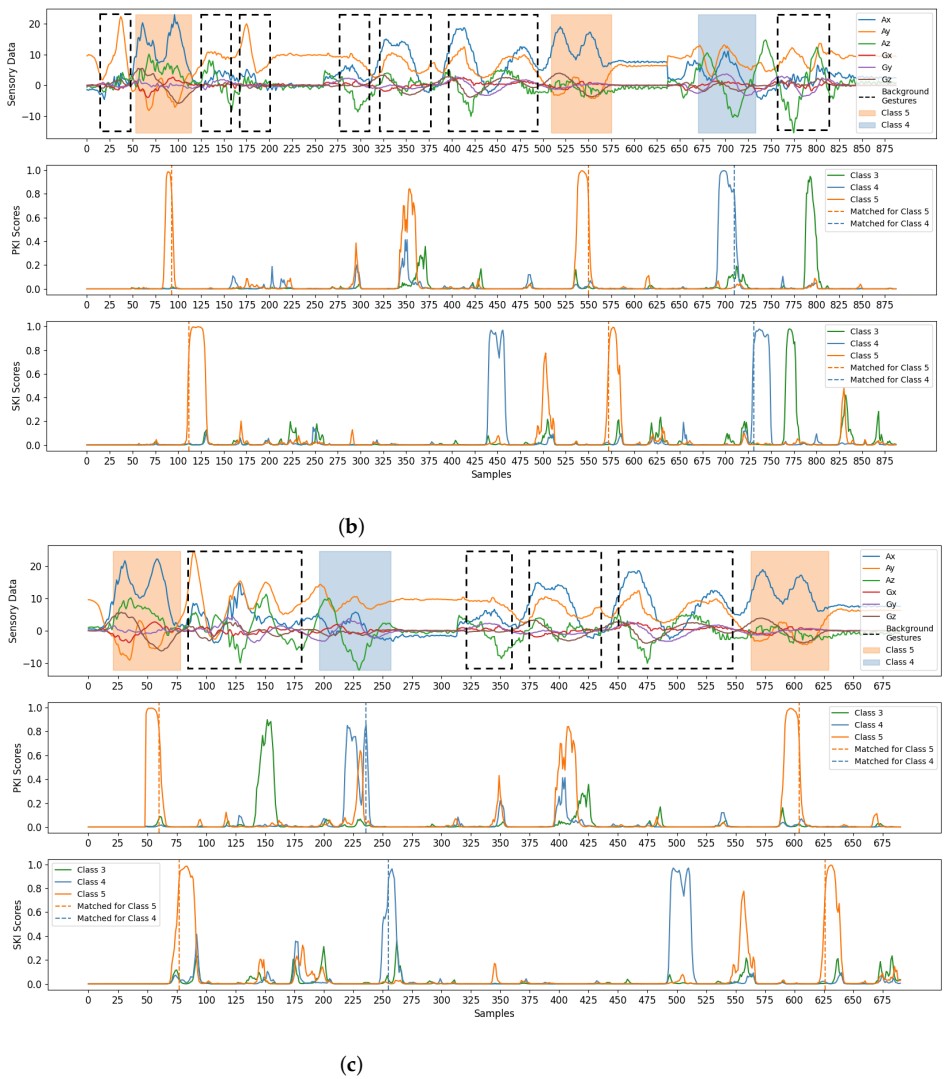

(**b**)

(**c**)

**Figure 10.** Examples for the detection and classification of the proposed algorithm. There are three test sequences considered in the example. Each sequence is a mixture of foreground gestures and background samples/gestures. The foreground gestures and background gestures are marked by coloured and dotted blocks, respectively. The PKI or SKI scores resulting in matched detections are marked by dotted lines. Each foreground gesture results in a pair of matched detections for PKI and SKI. Therefore, we are able to detect all the foreground gestures in the examples, (**a**) Example 1, (**b**) Example 2, (**c**) Example 3.

## 5. Concluding Remarks and Future Work

The proposed SBR algorithm was found to be effective for gesture detection and recognition. In our experiments, smart phones with accelerometers and gyroscopes were employed for the collection of sensory data for training and testing. It could be observed from the experiments that the proposed algorithm attained a high AUROC performance even for small training sets when the ITR ratio values were above 0.2. In addition, the proposed algorithm outperformed exiting object detection algorithms such as CornerNet in terms of AUROC for gesture detection. Furthermore, the proposed algorithm had superior classification accuracy over approaches such as SVM, LSTM, CNN, and Residual PairNet for gesture classification. The proposed algorithm also had a robust average EDA against the selection of thresholds $T$ in a large range for gesture detection. Even with a large stride size $\Delta = 4$ for the inference, the proposed algorithm was able to achieve an average EDA above 0.8. The algorithm therefore is an effective alternative for sensor-based HMI applications requiring both accurate detection and classification for hand gestures.

An extension of this work is the combination of SBR and VBR for human activity recognition (HAR), where gesture recognition can be considered as a special case. For many applications, it would be beneficial to achieve accurate HAR by exploiting sensors with different modalities, such as inertial or visual ones. While large body movements are differentiated by cameras, small hand actions can be captured by accelerometers. Larger varieties of actions can then be detected and/or classified with the presence of multiple and multimodal sensors. However, actions and/or gestures best-suited to specific applications could still be a challenging issue to be explored in the future.

**Author Contributions:** Conceptualization, Y.-L.C. and W.-J.H.; methodology, Y.-L.C. and T.-M.T.; software, Y.-L.C., T.-M.T. and P.-S.C.; validation, W.-J.H. and P.-S.C.; resources, W.-J.H.; writing—original draft preparation, W.-J.H.; writing—review and editing, W.-J.H. and T.-M.T.; visualization, Y.-L.C. and P.-S.C.; supervision, W.-J.H.; project administration, W.-J.H.; funding acquisition, W.-J.H. All authors have read and agreed to the published version of the manuscript.

**Funding:** The original research work presented in this paper was made possible in part by the Ministry of Science and Technology, Taiwan, under grants MOST 110-2622-E-003-003 and MOST 111-2622-E-003-001.

**Institutional Review Board Statement:** Not applicable.

**Informed Consent Statement:** Not applicable.

**Data Availability Statement:** The data are contained within the article.

**Conflicts of Interest:** The funders had no role in the design of the study; in the collection, analyses, or interpretation of data; in the writing of the manuscript, or in the decision to publish the results.

## Abbreviations

The following abbreviations are used in this manuscript:

| | |
|---|---|
| HAR | Human activity recognition |
| AUROC | Area under receiver operating characteristics |
| AP | Average pooling |
| Bi-LSTM | Bidirectional long short-term memory |
| CNN | Convolution neural network |
| CONV | Convolution |
| EDA | Edit distance accuracy |
| EMG | Electromyography |
| FPR | False positive rate |
| GN | Group normalization |
| HMI | Human–machine interface |
| IoT | Internet of Things |
| ITR | Key interval length to gesture radius |
| LSTM | Long short-term memory |
| PPG | Photoplethysmography |
| PKI | Primary key interval |
| ROC | Receiver operating characteristics |
| SBR | Sensor-based recognition |
| SKI | Secondary key interval |
| SVM | Support vector machine |
| TPR | True positive rate |
| VBR | Vision-based recognition |

## Appendix A. Frequently Used Symbols

**Table A1.** A list of symbols used in this study.

| | |
|---|---|
| $\Delta$ | The stride size for inference. |
| $A_t$ | The loss for PKI scores for all classes for input $S_t$. |
| $A_{t,j}$ | The loss for PKI score of class $j$ for input $S_t$. |
| $B_t$ | The loss for SKI scores for all classes for input $S_t$. |
| $B_{t,j}$ | The loss for SKI score of class $j$ for input $S_t$. |
| $C_t$ | The loss for classification scores for all classes for input $S_t$. |
| $C_{t,j}$ | The loss for classification scores of class $j$ for input $S_t$. |
| $E$ | Edit distance between two gesture sequences. |
| $F_t$ | The ground truth of PKI scores $X_t$. |
| $F_{t,j}$ | The $j$th element of $F_t$. It is the ground truth of $X_{t,j}$. |
| $G_t$ | The ground truth of SKI scores $Y_t$. |
| $G_{t,j}$ | The $j$th element of $G_t$. It is the ground truth of $Y_{t,j}$. |
| $H_t$ | The ground truth of classification scores $Z_t$. |
| $H_{t,j}$ | The $j$th element of $H_t$. It is the ground truth of $Z_{t,j}$. |
| $I$ | The length of a key interval. |
| $\mathcal{I}_R$ | The set of indices of samples in a gesture. |
| $\mathcal{I}_{\mathrm{PKI}}$ | The set of indices of samples in PKI of a gesture. |
| $\mathcal{I}_{\mathrm{SKI}}$ | The set of indices of samples in SKI of a gesture. |
| $K$ | The number of gesture classes for classification. |
| $L$ | The number of samples in the input sequence $S$. |
| $M_{\mathrm{TR}}$ | Size of training set. |
| $M_{\mathrm{TE}}$ | Size of testing set. |
| $N$ | The dimension of each sample $s_t$ in the input sequence $S$. |
| $P_s$ | The location of the starting sample of a gesture. |
| $P_f$ | The location of the ending sample of a gesture. |
| $Q_i$ | The classification accuracy for gesture class $i$. |
| $R$ | The radius of a gesture. |
| $r$ | ITR ratio defined in Equation (5). |
| $S$ | An input sequence to the proposed neural network. |
| $s_t$ | The $t$th sample in the input sequence $S$. |
| $S_t$ | A window in the input sequence $S$. The central sample of the window is $s_t$. |
| $T$ | Threshold value for the detection of PKI and SKI. |
| $U$ | The length of the ground truth of the gesture sequence (in number of gestures). |
| $u_c$ | The centroid of a gesture. |
| $u_{\mathrm{PKI}}$ | The centroid of PKI of a gesture. |
| $u_{\mathrm{SKI}}$ | The centroid of SKI of a gesture. |
| $W$ | Size of the window $S_t$. |
| $X_t$ | The PKI scores for $S_t$ by the proposed neural network. |
| $X_{t,j}$ | The $j$th element of $X_t$. It is the PKI score for class $j$. |
| $Y_t$ | The SKI scores for $S_t$ by the proposed neural network. |
| $Y_{t,j}$ | The $j$th element of $Y_t$. It is the SKI score for class $j$. |
| $Z_t$ | The classification scores for $S_t$ by the proposed neural network. |
| $Z_{t,j}$ | The $j$th element of $Z_t$. It is the classification score for class $j$. |

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
