# Peer review of "Sensor-Based Hand Gesture Detection and Recognition by Key Intervals"

_applsci, doi:10.3390/app12157410_

Round 1

Reviewer 1 Report

Review for MDPI of "Sensor-Based Hand Gestures ..."

5 July 2022, Paul C. Kainen

The achievement of the authors is the design of a network

which can detect and classify hand-gestures from sensor 

(not visual) data.  

The paper is good and is publishable in its current form.

Chen, Hwang, Tai, and Cheng use the

the fundamental idea of Caruana, whom they credit, of

Multitask Learning as a theoretical framework to give 

a specific branch structure and layering (and also algorithm) 

which are novel and evidently quite effective.  This is

the engineering and does not yet seem susceptible of

automation - perhaps just as well!  A similar example

in the past comes to mind: Sejnowski's NetTalk.

Having more layers, the superiority of Chen-et-al's

methods over CornerNet isn't particularly surprising.

Details are very nicely provided (ROC and AUROC). 

It would be interesting to try to quantitatively compare

accuracy and network resource cost using various recent

proposals such as representational and path norms. Perhaps

the functional analysis approaches of theoreticians

can provide pragmatic (and calculable) norms with

respect to which optimization can have better properties.

The paper under review is a very solid job of engineering

and offers, I think, a blueprint for further investigation. 

The sponsoring corporation might want to consider

combining the SBR tool with VBR algorithms.  And

eventually they should go the whole Oz Paradigm,

putting users and psychologists into the loop.

If there is any flaw in the paper, it is the gap

between detecting/classifying one in five gestures

and what would really be wanted in game situations

where a very nuanced gesture should produce a very

nuanced outcome - think of music for example, or of

a tennis player striking the ball with particular

direction and slice.

But such gap only reflects the actuality of our

current abilities with neural networks in practice:

strongly improving with a long way yet to go.

Author Response

We would like to thank the reviewers for the thorough review of the paper. The comments made by the reviewers are very useful, and we have revised the paper accordingly. All the revisions are marked in the revised paper. We hope the revisions are satisfactory.

“…The paper under review is a very solid job of engineering and offers, I think, a blueprint for further investigation. The sponsoring corporation might want to consider combining the SBR tool with VBR algorithms.  And eventually they should go the whole Oz Paradigm, putting users and psychologists into the loop.

If there is any flaw in the paper, it is the gap between detecting/classifying one in five gestures and what would really be wanted in game situations where a very nuanced gesture should produce a very nuanced outcome - think of music for example, or of a tennis player striking the ball with particular direction and slice.

But such gap only reflects the actuality of our current abilities with neural networks in practice: strongly improving with a long way yet to go.”

We thank the reviewer for the favorable review of the paper. We agree with the reviewer that the combination of SBR and VBR algorithms for Human Action Recognition (HAR) would be an important extension of the proposed work. Furthermore, the definitions of actions and/or gestures best suited to specific applications need to be explored in the future. The corresponding discussions have been included in the Section 5 (Concluding Remarks and Future Work) of the manuscript.

(Lines 366-393 on Page 17)

Reviewer 2 Report

I have a major concern on the motivation of sensor-based recognition. What's its advantage agains vision-based solutions? What's the particular use cases of sensor-based recognition. More detailed discussions about their difference are essential.

The experiments only demonstrate the effectiveness of the method on a limited number of gestures, which have very typical patterns. However, in true cases, the gestures may be more complex, and many gestures mix together. I am concerning whether the method will work well in real-world cases. 

The model shown in Fig. 2 is actually for multi-task learning. Some other multi-task methods on human-centric analysis like Differentiable multi-granularity human representation learning for instance-aware human semantic parsing should be reviewed.

Author Response

Reviewer #2

We would like to thank the reviewers for the thorough review of the paper. The comments made by the reviewers are very useful, and we have revised the paper accordingly. All the revisions are marked in the revised paper. We hope the revisions are satisfactory.

“I have a major concern on the motivation of sensor-based recognition. What's its advantage against vision-based solutions? What's the particular use cases of sensor-based recognition. More detailed discussions about their difference are essential.”

We thank the reviewer for this comment. Because Vision-Based Recognition (VBR) systems may record users’ life continuously, there are risks of personal information disclosure, which lead to privacy issues. The Sensor-Based Recognition (SBR) systems may be beneficial for the cases where privacy-preserved activity recognition is desired. Detailed discussions and corresponding references are included in the revised manuscript.

(Lines 22-26 on Page 1; Line 28-29 on Page 1; Ref. [2][3] on Page 19; Ref. [4] on Page 20)

“The experiments only demonstrate the effectiveness of the method on a limited number of gestures, which have very typical patterns. However, in true cases, the gestures may be more complex, and many gestures mix together. I am concerning whether the method will work well in real-world cases.“

In the revised manuscript, visual examples for gesture detection and classification are further provided, as shown in the Figure 10 of the revised manuscript. Each test sequence in the example is a mixture of foreground gestures and background gestures and samples. The test sequences are actually selected from the testing set adopted in the study. We can see from Figure 10 that, even with the presence of background gestures, the foreground gestures can still be effectively identified.

Please note that the background gestures defined in Figure 7 are the simple gestures constituting other unintended gestures in real life applications. Therefore, the avoidance of unexpected triggering of these background gestures is beneficial for accurate gesture detection and classification. All these results show the effectiveness of the algorithms.

(Figure 10 on Page 16; Lines 362-371 on Page 17)

“The model shown in Fig. 2 is actually for multi-task learning. Some other multi-task methods on human-centric analysis like Differentiable multi-granularity human representation learning for instance-aware human semantic parsing should be reviewed.”

We agree with the reviewer that the model shown in Figure 2 is actually for multi-task learning. As pointed out by the reviewer, multi-task training has also been found to be effective in human-centric analysis. The review of the corresponding examples such as Differentiable multi-granularity human representation learning for instance-aware human semantic parsing have been included in the revised manuscript.

(Line 106 on Page 3; Lines 108-112 on Page 3; Lines 135-136 on Page 4; Ref. [28][29][30] on Page 20)

Round 2

Reviewer 2 Report

The revision has addressed my concerns.